# A genome resource for *Acacia*, Australia's largest plant genus

Todd G. B. McLay[1,2,3☯]*, Daniel J. Murphy[1], Gareth D. Holmes[1], Sarah Mathews[3,4], Gillian K. Brown[5], David J. Cantrill[1], Frank Udovicic[1], Theodore R. Allnutt[1], Chris J. Jackson[1☯]

**1** Royal Botanic Gardens Victoria, South Yarra, Victoria, Australia, **2** School of BioSciences, The University of Melbourne, Parkville, Victoria, Australia, **3** Centre for Australian Biodiversity Research, CSIRO, Black Mountain, Australian Capital Territory, Australia, **4** Department of Biological Sciences, Louisiana State University, Baton Rouge, Louisiana, United States of America, **5** Queensland Herbarium, Department of Environment and Science, Toowong, Queensland, Australia

☯ These authors contributed equally to this work.
* todd.mclay@rbg.vic.gov.au

**Data Availability Statement:** Raw sequence and the assemblies are available on the Bioplatforms Data Portal (https://data.bioplatforms.com/organization/pages/bpa-plants/data-access), and NCBI (BioProject PRJNA752212). All bioinformatic

## Abstract

*Acacia* (Leguminosae, Caesalpinioideae, mimosoid clade) is the largest and most widespread genus of plants in the Australian flora, occupying and dominating a diverse range of environments, with an equally diverse range of forms. For a genus of its size and importance, *Acacia* currently has surprisingly few genomic resources. *Acacia pycnantha*, the golden wattle, is a woody shrub or tree occurring in south-eastern Australia and is the country's floral emblem. To assemble a genome for *A. pycnantha*, we generated long-read sequences using Oxford Nanopore Technology, 10x Genomics Chromium linked reads, and short-read Illumina sequences, and produced an assembly spanning 814 Mb, with a scaffold N50 of 2.8 Mb, and 98.3% of complete Embryophyta BUSCOs. Genome annotation predicted 47,624 protein-coding genes, with 62.3% of the genome predicted to comprise transposable elements. Evolutionary analyses indicated a shared genome duplication event in the Caesalpinioideae, and conflict in the relationships between *Cercis* (subfamily Cercidoideae) and subfamilies Caesalpinioideae and Papilionoideae (pea-flowered legumes). Comparative genomics identified a suite of expanded and contracted gene families in *A. pycnantha*, and these were annotated with both GO terms and KEGG functional categories. One expanded gene family of particular interest is involved in flowering time and may be associated with the characteristic synchronous flowering of *Acacia*. This genome assembly and annotation will be a valuable resource for all studies involving *Acacia*, including the evolution, conservation, breeding, invasiveness, and physiology of the genus, and for comparative studies of legumes.

## Introduction

*Acacia* Mill. is the largest genus of flowering plants in Australia, with 1,071 species (1,082 accepted species globally; http://worldwidewattle.com/infogallery/species/, accessed 6 July 2021). The diversification of *Acacia* in Australia represents a spectacular continent-wide

scripts and methods are available on GitHub (https://github.com/chrisjackson-pellicle/acacia_pycnantha_genome_manuscript).

**Funding:** Support for this study was provided by the Pauline Ladiges Plant Systematics Fellowship (Botany Foundation and Royal Botanic Gardens Victoria) in the form of funds to TGBM. The Genomics for Australian Plants Framework Initiative consortium provided support in the form of data. The funders had no role in study design, data analysis, decision to publish, or preparation of the manuscript.

**Competing interests:** The authors have declared that no competing interests exist.

radiation. Distributed in all ecosystems, with a particular richness in the arid and semi-arid biomes, *Acacia* extends from rainforest to alpine environments, forming a dominant component of many ecological communities [1, 2]. Phylogenetic dating and palynological fossil evidence support estimates that *Acacia* emerged *c.*23 Ma, and the diversification rate of the genus increased 15 Ma associated with climatic change in Australia [2]. Multiple clades of *Acacia* occur in all biomes, indicating repeated evolution of morphological characters and physiological adaptations associated with survival in a range of environments, including high levels of aridity, salinity, and alkaline soils [3, 4]. Phylogenetic analyses reveal several large clades that have rapidly radiated subsequent to the Pliocene, but the underlying evolutionary innovations that have driven this success are not fully understood [2, 5].

The significant morphological, physiological, and species diversity of *Acacia* represents substantial—and relatively untapped—genetic resources with potential for significant agricultural, environmental, and economic uses [6]. In tropical forestry, some species of *Acacia* form an important resource with over two million hectares of tropical Australian *Acacia* planted in south-east Asian countries for agro-forestry [7]. There has also been considerable use of *Acacia* species for land reclamation and agro-forestry, especially in areas affected by dry-land salinity [8]. Certain reproductive and physiological traits of a large number of *Acacia* species have contributed to their invasiveness in non-native habitats as their global use increased [9]. Species of *Acacia* are not currently widely used as a domesticated crop, but there has been limited selection of species for the use of seed as a food crop. This work has largely been guided by the traditional use of mostly arid-adapted species by Indigenous Australians [10]. While the commercial potential of *Acacia* is still being developed, the environmental and physiological diversity within the genus suggests *Acacia* will play a significant role as we adapt to a changing climate [11].

*Acacia* is a member of the nitrogen-fixing legume family Leguminosae, which is the third largest family of angiosperms and is regarded as the second-most economically important family after Poaceae. *Acacia* belongs to subfamily Caesalpinioideae in the informally named 'mimosoid clade' [12, 13]. Overall, the Leguminosae are well represented by genomic data, with assembled genomes for species including soybean (*Glycine max*), chickpea (*Cicer arietinum*) and peanut (*Arachis hypogaea*). However, taxonomic representation in these data remains distinctly biased towards the largest subfamily, Papilionoideae (see review in 12). Commonly known as the "pea-flowered" legumes, Papilionoideae contains many crop species; genomic work has focussed mostly on this clade of legumes due to the potential for economic benefits. Given the relative dearth of genomic data for other legume subfamilies, and especially for mimosoid species, a genome resource for *Acacia* is particularly valuable; for comparative studies of important plant traits across all legume subfamilies, better representation of Caesalpinioideae has been critical.

An *Acacia* genome is a strategic resource for the study of genomic adaptations leading to the continent-wide success of the genus, and subsequently for advancing our understanding of the evolution of the Australian flora and its biomes. It is also a key resource for conservation genomics of species of *Acacia* [14, 15], invasion genomics [16], ethnobotany [17], and forestry [18]. In this study, we use long-read (Oxford Nanopore—ONT), linked read (10X) and Illumina short-read sequencing technologies to assemble the genome of *Acacia pycnantha*, Australia's official floral emblem (http://www.anbg.gov.au/emblems/aust.emblem.html, accessed July 2021; Fig 1).

## Material and methods

### Bioinformatic analyses: Commands and scripts

For details of bioinformatic commands, settings, and scripts, see the GitHub repository at https://github.com/chrisjackson-pellicle/acacia_pycnantha_genome_manuscript.

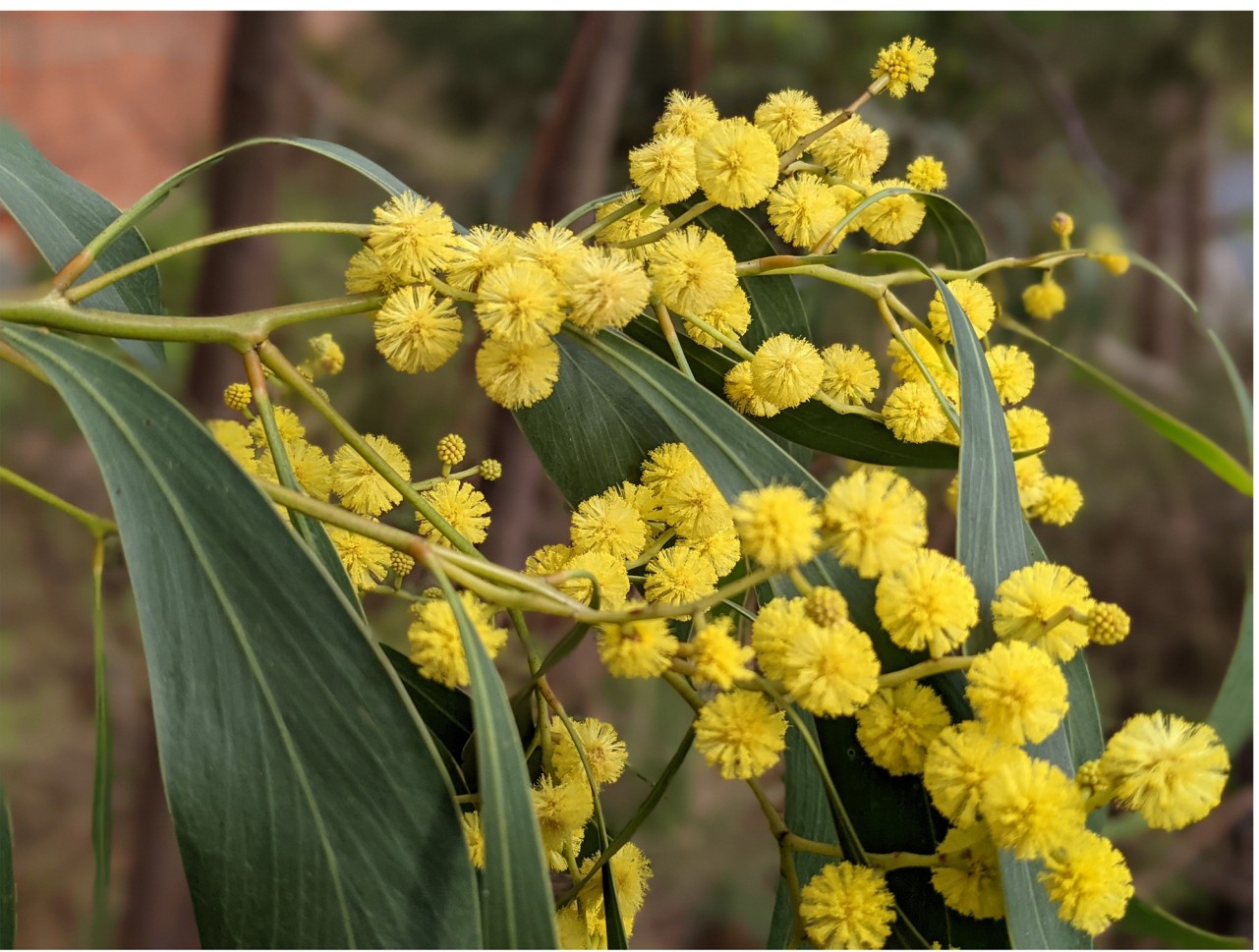

**Fig 1. *Acacia pycnantha*, showing inflorescences and phyllodes (naturalised on Phillip Island, Victoria, Australia; photo: Dan Murphy).**

## Plant materials, DNA extraction, and flow cytometry

Young phyllodes, buds, and fruit were collected from a plant growing at the Australian National Botanic Gardens (voucher details: CANB 748486.1—S.R. Donaldson 3550 12/10/2007). The original provenance of the seed collection of *Acacia pycnantha* was the Warby Ranges, in north-eastern Victoria (E.Canning 3243). Phyllodes were used for DNA extractions on the day of collection, and young phyllodes, buds, and fruits were also placed in RNAlater (Sigma-Aldrich) for preservation. High molecular weight DNA was extracted using a modified CTAB protocol with a sorbitol prewash (Inglis et al 2018, see link for full extraction protocol https://www.genomicsforaustralianplants.com/wp-content/uploads/2020/03/DNA-extraction-Acacia-pycnantha.pdf). The genome size of the *A. pycnantha* plant used for genome assembly was estimated by flow-cytometry using CyStain PI Absolute P (Sysmex Partex GmbH, Görlitz, Germany), and *Zea mays* as an internal standard. The sample was measured with a 488 nm laser (BD Accuri C6 Plus equipped with a BD CSampler Plus, BD Biosciences, San Jose, CA, USA) and run at a flow rate of 14 µm/min and core size of 10 µm. Histogram data were collected using the FL2 detector while eliminating events with a value of less than 5000 on FL2-H. Analysis was performed with the BD Accuri C6 Software version 1.0.23.1.

## Genome sequencing and filtering

For ONT sequencing, approximately 10 μg of high molecular weight (HMW) DNA was size-selected using a BluePippin (Sage Science) to remove DNA fragments less than 10 kb and sequenced using MinION and PromethION (Oxford Nanopore) devices (see S1 Table in S2 File for flowcell and base-calling software versions). ONT reads were visualized and assessed using tools in the NanoPack package [19]. FASTQ reads were pooled and assessed using Nano-Plot v1.24.0 (https://github.com/wdecoster/NanoPlot). Pooled reads were subsequently filtered using NanoFilt v2.3.0 (https://github.com/wdecoster/nanofilt) and again assessed with Nano-Plot. Filtered reads were self-corrected using the correction stage of the Canu v1.9 assembler [20] (see S2 Table in S2 File for details).

An aliquot of the size-selected DNA was also prepared for Chromium 10X Genomics linked-read sequencing following the manufacturer's protocol for library preparation. The 10X barcoded library was sequenced using Illumina sequencing technology (NovaSeq 6000). To generate barcode-attached reads in FASTQ format, read data was processed using the LongRanger 2.2.2 basic pipeline (10X Genomics) with default settings.

For Illumina short-read sequencing, genomic DNA libraries were prepared using the Illumina TruSeq Nano workflow with 100 ng of input DNA that was mechanically fragmented to 350 bp insert size prior to preparation and. The library was sequenced using a NovaSeq 6000 with 150 bp paired-end (PE) reads. Reads were trimmed and filtered using BBduk from the BBMap v38.61 software suite (S3 Table in S2 File). Trimmed and filtered reads were used to estimate genome size with JellyFish version 2.3.0 [21] and GenomeScope version 1.0.0 [22].

Total RNA was extracted using a modified NucleoSpin RNA Plant and Fungi Kit (Macherey-Nagel, Germany), following a sorbitol clean for the flower and fruit material (see https://www.genomicsforaustralianplants.com/wp-content/uploads/2020/06/RNA-extraction-Acacia-pycnantha.pdf for detailed methods). RNA libraries for each of the three tissues were prepared using the Illumina TruSeq Stranded mRNA workflow with an insert size of 150–180 bp and sequenced on a NovaSeq 6000 with 150 bp PE reads.

Reads were trimmed and filtered using Trimmomatic v0.39 [23]. For structural gene annotation hints, filtered reads were normalised to ~100× coverage using BBNorm from the BBMap v38.44 software suite (https://sourceforge.net/projects/bbmap/).

## Long read genome assembly, polishing and scaffolding

To identify a suitable genome assembly approach we tested multiple assembly programs, including long-read only assemblers (NECAT [24], Canu v2.2 [20], Flye v2.6 [25], Wtdbg2 v2.5 [26]) and hybrid assemblers that use both long ONT and short Illumina reads (HASLR v0.8a1 [27], WenganD v0.1 [28]; see S4 Table in S2 File). The program NECAT was selected as the best candidate and used to assemble raw Nanopore data (see S1A in S1 File for configuration details), generating 2,323 contigs totaling 1,069,632,449 bases with an N50 of ~962 kb (see S5 Table in S2 File for further details). Subsequently, three rounds of long-read polishing were performed using the filtered, corrected Nanopore reads, once with Racon v1.3.3 [29] followed by two rounds with Medaka v0.11.5 (https://github.com/nanoporetech/medaka). Medaka splits contigs at positions where no reads span a region of the draft sequence, as reflected in the increased number of contigs shown in S5 Table in S2 File. Finally, a round of short read polishing was performed with Racon using the trimmed and filtered Illumina data (forward reads only at ~55× coverage) with default settings.

To identify and split potentially misassembled contigs, the polished contigs were processed using Tigmint v1.1.2 [30] with the 10X linked-read data output from the LongRanger basic pipeline, using default settings. Then, purge_dups v0.0.3 [31] was used to remove haplotigs

and heterozygous contig overlaps, using both forward and reverse filtered Illumina shotgun reads and filtered Nanopore reads.

Contigs were scaffolded using filtered, corrected Nanopore reads with RAILS v 1.5.1 /Cobbler v0.6.1 [32]. Scaffolded and gap-filled contigs were subsequently polished with two rounds of Racon using filtered, corrected Nanopore reads as described above, followed by two rounds of Racon using Illumina sequence data as described above. To split any potentially erroneous contig joins introduced by RAILS, polished contigs were again processed with Tigmint as described above. Finally, contigs were scaffolded using 10X linked-read data with ARCS v1.1.0 [33]. Contigs smaller than 1000 bp were excluded from further analysis. In addition, a script to calculate Shannon's entropy (kmercount-shannons.py, see GitHub repository) was used to identify four contigs larger than 1000 bp that consisted only of simple repeats which were also excluded.

## Genome quality control and completeness

For each stage of the genome assembly, statistics were generated using the software assembly-stats (https://github.com/sanger-pathogens/assembly-stats). Assembly quality was assessed using the k-mer spectrum of the filtered Illumina shotgun data with Merqury version 1.1 [34]. The k-mer database required by Merqury was generated using meryl [35] (version included with Canu version 2.2) with k = 21 (see S1B in S1 File for Merqury results and spectra plot).

## Repetitive elements annotation

A non-redundant transposable element (TE) library was generated using the EDTA pipeline version 1.8.4 [36]. To assist in filtering out gene-related sequences from the final TE library, EDTA was provided with nucleotide transcript sequences from the closely related taxon *Prosopis alba* (see NCBI BioProject accession PRJNA534081). The TE library output contained 12,447 sequences. TEs were then classified using the Transposon Classifier "RFSB" tool from TransposonUltimate version 1.0 [37], with the option [-mode classify]. Custom Python scripts were used to relabel the EDTA TE library sequences with the TransposonUltimate classification, with the following amendment: in cases where the TransposonUltimate classification probability of a sequence to either Class I (retrotransposons) or Class II (DNA transposons) was less than 0.5, the sequence was labelled as 'unclassified'. The relabelled TE library was used to soft-mask the genome assembly using RepeatMasker version 4.1.0 [38], and the output file produced by RepeatMasker was used to generate an annotation table (see S6 Table in S2 File) using the RepeatMasker script buildSummary.pl.

## NUPT and NUMT identification

To identify NUPTs (nuclear plastid DNA), NUMTs (nuclear mitchodrial DNA) and NUMPTs (loci that contain both NUPTs and NUMTs) in the genome assembly, the *A. pycnantha* plastome and mitome [39] were used as a query in BLAST searches of each nuclear scaffold. BLAST hits were filtered to include only those with an alignment length greater than 100 bp and with a minimum identity of 85%. Nested hits were removed, retaining only the longest contiguous hits. It is possible for NUMPTs to arise from assembly errors rather than genuine insertions into the nuclear genome (Shi et al., 2017). We considered NUMPTs which had Illumina reads mapped across their organelle DNA—nuclear DNA junction to be 'confirmed NUMPTs' and those that showed no overlap reads to be 'unconfirmed'. Illumina reads were mapped to scaffolds containing NUMPTs using BBMap (v38.61) and custom Python scripts (see git repository) were used to identify and count junction-mapped reads. An InterProScan

[40] gene annotation (see annotation methods below) was then used to identify confirmed NUMPTs which occurred within genes, and/or contained annotated genes within them.

## Gene prediction and functional annotation

Structural gene annotation of the TE-masked genome assembly was performed using the BRAKER2 pipeline [41]. ETP-Mode was used, which accepts evidence hints in the form of spliced RNAseq alignments and spliced protein alignments. To generate RNA-seq spliced alignment hints, we combined our quality filtered, 100× coverage Illumina RNAseq data with *Acacia pycnantha* RNAseq data available from the 1KP initiative [42] (NCBI BioProject accession PRJEB4922) and aligned the reads to our soft-masked genome using STAR [43]. The resulting BAM file was supplied to BRAKER2. To generate a database of proteins for BRAKER2 input, we filtered the OrthoDB v10.1 [44] catalog of orthologous protein-coding genes for Viridiplantae sequences (NCBI taxon ID 33090) and supplied the filtered protein families to BRAKER2. To remove putative transposons from this gene set (i.e., those that were not identified with the EDTA pipeline described above), Pfam domains were identified in the corresponding gene nucleotide sequences, and corresponding domain text descriptions were extracted from the Pfam website (http://pfam.xfam.org/). For each gene, Pfam descriptions were searched against a list of transposon-related terms (transcriptase, transposase, gag, env, transposon, repetitive element, RNA-directed DNA polymerase, pol protein, non-LTR retrotransposon, mobile element, retroelement, retrovirus, Retroviral, group-specific antigen). Where more than half of the Pfam domains in a gene had matches to one of these terms, the gene was flagged as a potential transposon and removed from the BRAKER2 predicted gene set. Finally, any gene that has no external support (i.e., RNAseq or OrthoDB protein alignment evidence) during BRAKER2 gene prediction, and also lacked a functional annotation (see below), was removed. See the GitHub repository for full methods.

Completeness of the resulting predicted protein-coding gene set was assessed using BUSCO v4.0.4 searching against both the embryophyta_odb10 (1,375 genes) and fabales_odb10 (5,366 genes) databases.

The predicted genes were assigned functions using four methods. Firstly, Pfam domains for each of the 15 angiosperm taxa were determined by searching each protein dataset against v33.1 of the PfamA.hmm database [45] using the hmmsearch program from HMMER v3.2.1 [46]. Secondly, amino-acid sequences corresponding to the filtered BRAKER2 predicted gene set (47,624 genes) were functionally annotated using eggNOG mapper v2 [47] with version 5.0 of the eggNOG database via the web portal (http://eggnog-mapper.embl.de/), see S7 Table in S2 File. Thirdly, KEGG Orthology (KO) annotation of the filtered BRAKER2 predicted gene set was performed using the BLAST algorithm implemented in BlastKOALA [48] via the KEGG website (https://www.kegg.jp/blastkoala/), see S8 Table in S2 File. Finally, the filtered BRAKER2 predicted gene set was annotated using InterProScan version 5.50–84.0 (see S3 File). A Venn diagram to compare the genes functionally annotated by each methodology was produced using TbTools [49].

## Identification of orthologous gene families

To compare the diversity and abundance of *A. pycnantha* gene families to other species of legumes and angiosperms more broadly, gene families (orthogroups) were calculated using OrthoFinder v2.3.12 [50]. Seven Leguminosae species including *A. pycnantha* were included in the analysis, along with eight other angiosperms (see S9 Table in S2 File); protein sets containing a single isoform for each gene were used. A corresponding species tree was generated based on APG IV [51] and established relationships between the Leguminosae genera [52]

(S1C, S1 Fig in S1 File). OrthoFinder was run using default settings with the species tree provided as input. A second OrthoFinder run was also performed using only the seven Leguminosae protein sequences (S1C, S2 Fig in S1 File). Visualisations of selected OrthoFinder results were generated using a modified version of the script Fig 1_ResultsOverview.py, originally available at https://zenodo.org/record/1481147#.X5ognVlxXUI, see also the GitHub repository for this study.

## Analyses of genome evolution

A chronogram was generated for the 15 angiosperm taxa using relaxed molecular clock methods implemented in PhyloBayes v4.1b [53]. For input sequence data, single gene amino-acid alignments were generated from 85 single-copy orthogroups (SCO) identified in the angiosperms OrthoFinder analysis, using the MUSCLE algorithm [54]. Each alignment was manually trimmed to remove poorly aligned regions in Geneious Prime 2020 (BioMatters, New Zealand), and trimmed alignments were concatenated to generate a supermatrix 40,206 amino acids in length (S4 File). The species tree generated for OrthoFinder analysis was provided as a fixed topology. Calibration points were provided for seven nodes (S10 Table in S2 File), using time-range estimates recovered from TimeTree [55], http://www.timetree.org/). To better account for changes in rates of molecular evolution throughout the angiosperms, PhyloBayes was run using the uncorrelated gamma multiplier model, global exchange rates were inferred from the data and 4 gamma categories were used. Two Markov chain Monte Carlo (MCMC) were run in parallel for ~16,240 cycles each, and convergence of likelihoods and parameter estimates was assessed in Tracer 1.7 [56]. For the final chronogram, chain 1 was summarized using the readdiv program, discarding the first 7,500 cycles as burn-in based on the chain convergence profile (see S1D, S3 Fig in S1 File).

To explore gene-tree/species-tree concordance, a maximum likelihood tree was generated from a concatenated alignment of the 85 SCOs in IQTREE [57] allowing each SCO to have an estimated substitution rate (-m TEST). Support for the topology was estimated using 1000 UFBoot replicates using BNNI correction, and SH-aLRT was used as an independent test of branch support [58]. Gene trees for each SCO were also estimated, and gene concordance factors and site concordance factors were mapped on to the concatenated alignment phylogeny with 100 quartet replicates [59].

Whole genome duplication tests were performed with the software wgd [60]. The 47,624 *A. pycnantha* predicted gene CDS sequences were filtered to remove any sequence not starting with a canonical ATG start codon or with a length that was not a multiple of three, leaving 47,460 sequences. To obtain the *A. pycnantha* 'paranome' (the collection of paralogous genes), an all-vs-all BLASTp was performed followed by clustering with MCL [61] via wgd. The paranome $K_S$, $K_A$ and $\omega$ distributions were calculated using the default aligner MAFFT [62] and the default phylogenetic tree reconstruction program FastTree [63]. Anchor pairs were identified using the *A. pycnantha* GFF file from the BRAKER2 annotation. Mixture models were estimated using both the gmm and bgmm methods. Finally, the $K_S$ distribution histogram and Kernel Density Estimations were visualised. The same analysis was subsequently performed on the Leguminosae taxa *Prosopis alba*, *Senna tora*, *Cercis canadensis* and *Lupinus angustifolius* to identify duplication events throughout the clade; as for OrthoFinder analyses, analyses were performed using gene sets containing a single isoform per gene.

## Gene family evolution

Gene family expansions and contractions within the Leguminosae were estimated using CAFÉ v5.0 [64]. CAFÉ analyses require that each gene family has at least one gene at the root of the

tree; gene families failing this criterion are filtered out prior to analysis. Therefore, to ensure that as many *A. pycnantha* gene families as possible were analysed, we performed CAFÉ analyses with a pruned chronogram comprising Leguminosae taxa only, along with a gene-count table derived from the Leguminosae-only OrthoFinder analysis (S11 Table in S2 File); for full methods see S1E in S1 File.

To identify Pfam domains or gene families that are significantly expanded or reduced in *A. pycnantha* compared to other angiosperms, we calculated z-scores for each Pfam entry. Per-species Pfam domain counts were generated, and z-scores were calculated; domains with a z-score > 1.96 or < -1.96 in *A. pycnantha* were considered significantly expanded or contracted, respectively (see S12 Table in S2 File; see GitHub repository for full methods and scripts). Alignments were produced for gene families of interest, poorly aligned positions were removed using Gblocks [65], and phylogenies were generated using IQTREE.

GO-term enrichment analyses were performed on several *A. pycnantha* gene sets, based on GO terms assigned by InterProScan. Tested sets included: significantly expanded genes identified from CAFÉ and Pfam analyses, *Acacia*-only orthogroups, and single *Acacia* sequences that were not assigned to an orthogroup. GO enrichment analyses were performed using GOATOOLS [66] implementing the hypergeometric means test.

Finally, to determine whether any of the expanded or contracted gene sets, *Acacia*-only orthogroups, or unassigned *Acacia* genes identified above were enriched for specific KEGG pathways, we performed hypergeometric means tests. Overestimation of significant p-values was corrected using false-discovery rate correction. Pathways with a p-value of less than 0.05 were considered significantly enriched.

## Results and discussion

### Genome sequencing and assembly of the *Acacia pycnantha* genome

To produce a draft genome for *Acacia pycnantha*, we generated approximately 500 Gb of raw genomic sequence data. After quality filtering 283 Gb remained, comprising 109 Gb of Illumina NovaSeq shotgun sequencing, 35 Gb of Oxford Nanopore long read data (N50 read length of 12 kb), and 138 Gb of Illumina 10X linked-reads. In addition, 88 Gb of RNAseq Illumina sequence data were produced. The haploid genome size of *Acacia pycnantha* was estimated to be 0.6 Gb using GenomeScope, with an estimated heterozygosity of 0.61% (including a repeat length of 273 Mb, see S13 Table in S2 File). In comparison, genome size estimations using flow cytometry indicated a haploid genome size of 0.85 Gb (also see [67]). Based on the genome size estimated using flow-cytometry, the initial long-read assembly was performed using ~40× coverage of Nanopore long-read data. The draft genome is available under NCBI BioProject accesssion PRJNA752212.

The initial long-read assembly was ~1.0696 Gb with an N50 of 0.962 kb. After polishing the assembly with short-read Illumina data and filtered, corrected long-read Nanopore data, potentially misassembled scaffolds were split using 10X Chromium data. Haplotigs and heterozygous overlaps were removed, reducing the total assembly size to ~0.8187 Gb. Subsequent scaffolding and gap-filling with long-read Nanopore data increased the assembly N50 to ~1.383 Mb with a total length of ~0.8209 Gb. Following additional short and long-read polishing, 10X Chromium data was again used to split potentially misassembled scaffolds, and a final scaffolding stage was carried out also using Chromium data. The final scaffold set consisted of 1,267 scaffolds totaling ~0.8144 Gb with an N50 of ~2.8 Mb (Table 1, see S14 Table in S2 File for a comparison with the other Leguminosae genomes used in this study). This genome size is closer to the flow cytometry estimate (0.85 Gb) than the GenomeScope estimate (0.6 Gb). Genome size estimations using k-mer counting are known to be sensitive to features of the

**Table 1. Genome assembly and annotation statistics of the *Acacia pycnantha* genome.**

|  | *A. pycnantha* GENOME |
|---|---|
| **Genome Assembly Size (Mb)** | 814.40 |
| **G+C Content (%)** | 36.1 |
| **Number Of Scaffolds** | 1,267 |
| **Scaffold N50 (kb)** | 2,821 |
| **Scaffold L50 (number)** | 75 |
| **Number Of Contigs** | 1,697 |
| **Contig N50 (kb)** | 1,331 |
| **Contig L50 (number)** | 169 |
| **Number Of Ns** | 42,695 |
| **BUSCO (Genome)** | |
| EMBRYOPHYTA | C:95.8%[S:83.1%,D:12.7%],F:0.9%,M:3.3%; n:1375 |
| **Protein Coding Genes** | 47,624 |
| **BUSCO (Proteome)** | |
| EMBRYOPHYTA | C:98.3%[S:85.7%,D:12.6%],F:1.2%,M:0.5%; n:1375 |
| FABALES | C:90.5%[S:70.8%,D:19.7%],F:0.7%,M:8.8%; n:5366 |
| **Transposable Elements** | 62.26% |
| DNA TRANSPOSON | 1.99% |
| DNA TRANSPOSON/TIR | 18.43% |
| RETROTRANSPOSON/LTR | 37.39% |
| RETROTRANSPOSON/NON-LTR | 0.74% |
| UNCLASSIFIED | 3.71% |

genome such as high repetitiveness and/or heterozygosity [68]. It is therefore not surprising that the two methods differ, as the *Acacia* genome appears to have a high level of repetitive DNA (see below) and has an estimated heterozygosity of 0.61%.

## *Acacia pycnantha* genome characterisation, annotation, and gene family clustering

Transposable elements (TEs) comprised ~62% of the total genome sequence (Table 1, S6 Table in S2 File). Most of the transposable elements belonged to long terminal repeat (LTR) retrotransposons (37.3% of the total genome, with 23.43% classified as Gypsy type LTRs), followed by DNA transposable elements (18.43%). The proportion of TEs in the *A. pycnantha* genome was the highest of any sequenced Leguminosae genome to date (S15 Table in S2 File).

We identified 179.3 kb of confirmed NUMTs (confirmed by junction Illumina read overlaps, see Methods), 438.3 kb of confirmed NUPTs, and 192.6 kb of confirmed NUMPTs (insertions containing both mitochondrial and plastid DNA). Unconfirmed insertions totalled: NUMTs, 149.7 kb; NUPTs, 71.8 kb; and NUMPTs, 327.6 kb (S16 Table in S2 File). The BLAST percent identity to the mitome and plastome references of confirmed and non-confirmed insertions was compared by ANOVA and found to be significantly lower in confirmed insertions (confirmed = 92.9%; non-confirmed = 96.8%; P = 1 x $10^{-25}$; S17 Table in S2 File). This would be expected if the confirmed insertions were genuine and more diverged from the organellar genomes than non-confirmed insertions, which may have arisen from assembly errors (and their divergence being predominantly due to sequencing errors). NUPT, NUMT, and NUMPT loci positions are given in S5 File for confirmed and non-confirmed insertions respectively (GFF3 format).

Transfer of organellar DNA to the nuclear genome is common in plants, and it can lead to structural and organisational variation in the genome [69, 70]. Transfers typically begin as large fragments near centromeres, and these are gradually broken up and shuffled around the genome by TEs [71]. However, apparent transfers can be caused by misassemblies where portions of the chloroplast and/or mitochondrial genomes are mistakenly incorporated into nuclear contigs [72, 73]. Our method utilising Illumina paired reads to test the assembly / insertion junctions identified that approximately 12% by number (40% by length) could not be confirmed as true NUMPTs. Manual examination of Illumina and ONT reads mapped to insertion sites showed that ONT reads alone carried the spurious insertions, possibly as a result of chimeric reads as previously reported [74, 75]. Although the ONT sequencing performed here did not involve a PCR step, chimeric reads may have arisen from an unknown process resulting from the large proportion of organelle DNA present in plant cells. We suggest that all putative organelle DNA insertions in plant genome assemblies arising from ONT reads should be tested with Illumina (or other short read methods) mapping to insertion / nuclear DNA junctions as performed here, because in isolation, chimeric ONT reads cannot be distinguished from real organelle DNA nuclear insertions. Further investigation of how such ONT chimeras have formed should also be undertaken. NUMPTs are not commonly checked in genome assemblies, or they are laboriously checked using PCR. Here, we provide a bioinformatic method to determine and investigate the source of NUMPTs in genome assemblies. Accurate identification of NUMTs and NUPTs is important for genome assemblies, and for understanding their role during evolution [76, 77].

Nuclear gene models were predicted using the BRAKER2 pipeline, followed by additional filtering to remove putative TEs and genes with little or no support (see Methods). In total, 47,624 genes remained after filtering. This number is comparable to most other Leguminosae (S9 Table in S2 File). Of the 47,624 predicted genes, 44,889 (94.2%) were functionally annotated by at least one source (eggNOG = 90.3%; InterProScan = 90.3%; KEGG = 30.3%; Pfam = 70.4%; Fig 2).

The completeness of the predicted proteome was assessed using BUSCO analyses. The predicted gene set contained complete sequences for 98.3% of the 1,375 Embryophyta BUSCO genes, with only 0.5% missing entirely, and complete sequences for 90.5% of the 5,336 Fabales BUSCO genes, with 8.8% of genes missing entirely (S18 Table in S2 File for full BUSCO results and a comparison with other Leguminosae taxa used in this study). The Caesalpinioideae and *Cercis* have less than 92% of the Fabales BUSCO genes, whereas the two Papilionoideae (*Glycine max*, *Lupinus angustifolius*) have greater than 97% of the Fabales BUSCO genes. This may indicate a proportion of the Fabales BUSCO gene set are specific to the Papilionoideae and should be taken into consideration when determining the completeness of other Leguminosae genomes.

OrthoFinder analysis of a set of 15 broadly sampled angiosperm species assigned 43,999 (92.4%) of the *A. pycnantha* genes to one of the 30,061 identified orthogroups (Fig 3, S19, S20 Tables in S2 File). A total of 34,713 (72.9%) *A. pycnantha* genes were present in an orthogroup containing an ortholog from at least one other angiosperm species. *Acacia pycnantha* had the highest proportion of species specific orthogroups, with 5,645 (11.8%) genes present in one of the 1,438 orthogroups containing *A. pycnantha* sequences only; 3,641 predicted *A. pycnantha* genes that did not have any identified orthologs. The Leguminosae-specific OrthoFinder analysis assigned 43,549 (91.4%) of the *A. pycnantha* genes to one of the 27,228 identified orthogroups (S21, S22 Tables in S2 File). A total of 32,251 (67.7%) *A. pycnantha* genes were present in an orthogroup containing an ortholog from at least one other Leguminosae species. *Acacia pycnantha* again had the highest proportion of species specific orthogroups, with 7,207 (15.1%) genes present in one of the 1,759 orthogroups containing *A. pycnantha* sequences

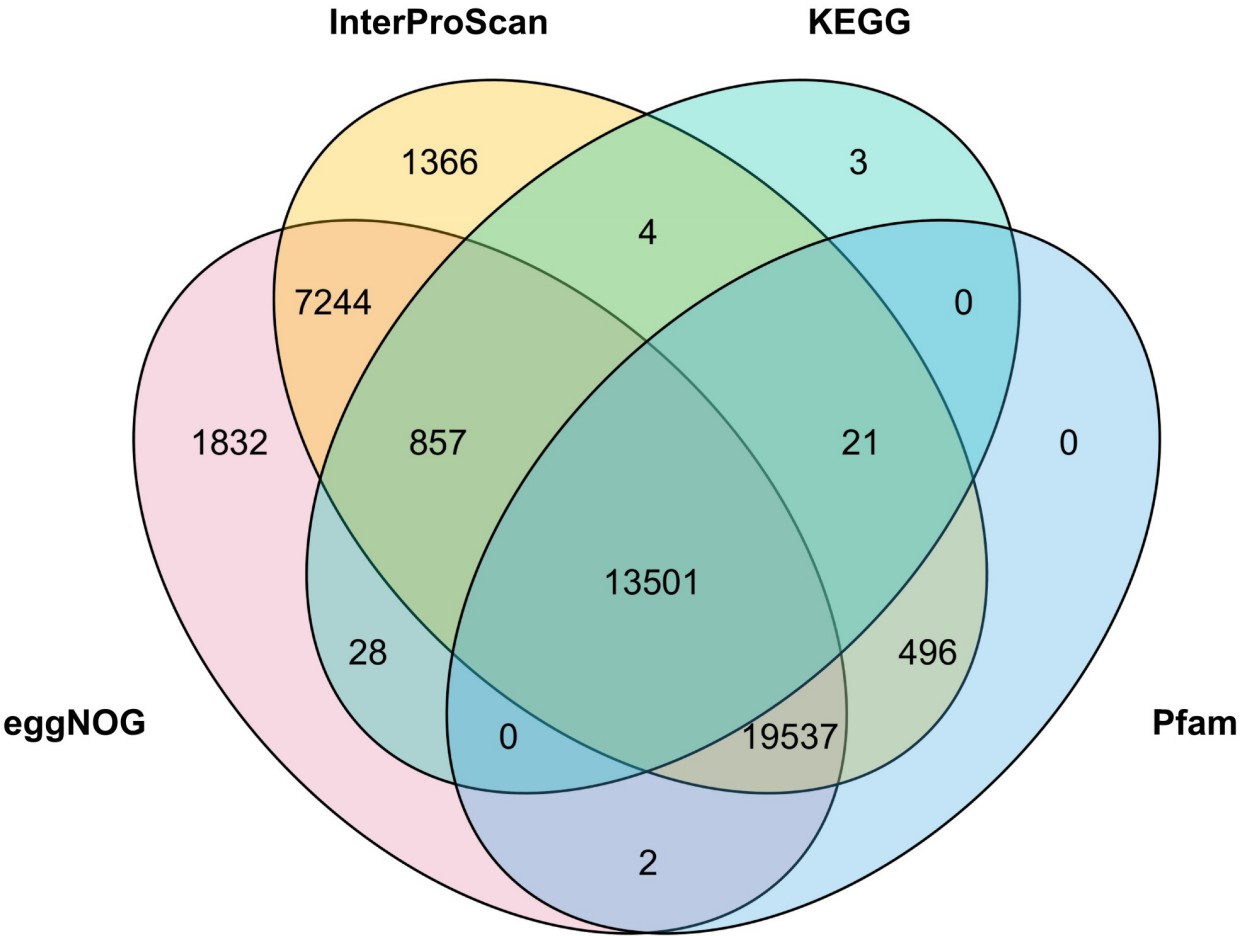

**Fig 2. Functional annotations Venn diagram comparing the overlap of the *Acacia pycnantha* proteome annotated using Pfam, eggNOG, KEGG, and InterProScan.**

only. There were 4,091 predicted *A. pycnantha* genes that did not have any identified orthologs.

## Evolutionary analyses

Phylogenetic dating estimated that *Acacia* and *Prosopis* diverged around 31 Ma (~24 Ma to 47 Ma 95% Highest Posterior Density (HPD), Fig 4). This is comparable to results from Koenen et al. [52], who estimated a divergence time of 33.9 Ma–34.4 Ma. These latter dates are within the 95% HPD of the estimate found in our analyses, and the minor difference is likely due to different taxonomic and gene sampling, as well as differing calibration points (Koenen et al. 2020 used a lower bound of 33.9 Ma on the *Acacia*/*Prosopis* node).

Whole genome duplications (WGD) are an important driver of plant evolution [78]. Multiple genome duplication events have been hypothesised in the Leguminosae, although their exact timing and placement has been difficult to ascertain due to the rapid diversification of the family into subfamilies, and genetic processes such as fractionation and diploidisation that obscure duplication events [79]. We used Kernel Density Estimates of $K_S$ distributions from one-to-one orthologs and anchor-pair paralogs to estimate speciation events and shared duplication events, respectively, between *Acacia* and four other Leguminosae taxa (S1F, S5-S7 Figs

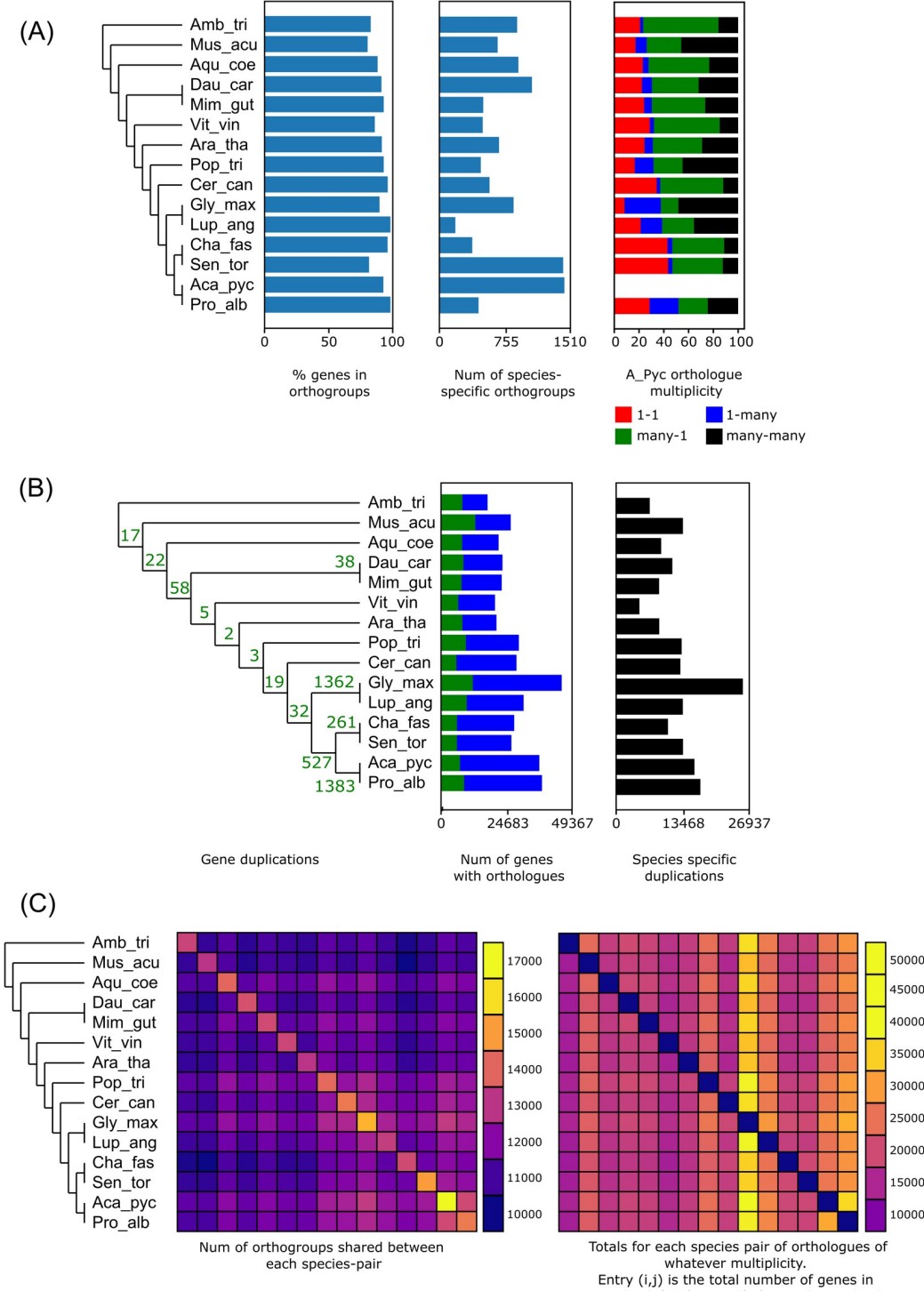

**Fig 3. OrthoFinder statistics from broad angiosperms sampling.** (A) Genes in orthogroups, number of species specific orthogroups, and ortholog multiplicity of all samples relative to *A. pycnantha*. (B) Estimated gene duplications on phylogeny, genes with orthologues, number of species-specific orthogroups. (C) Orthogroup overlap between species pairs. On-diagonal values in the left panel correspond to the total number of orthogroups present for each species. On-diagonal values in the right panel all equal zero; note that this heatmap is not a mirror image, as species *i* might have many more copies of a given ortholog than species *j*.

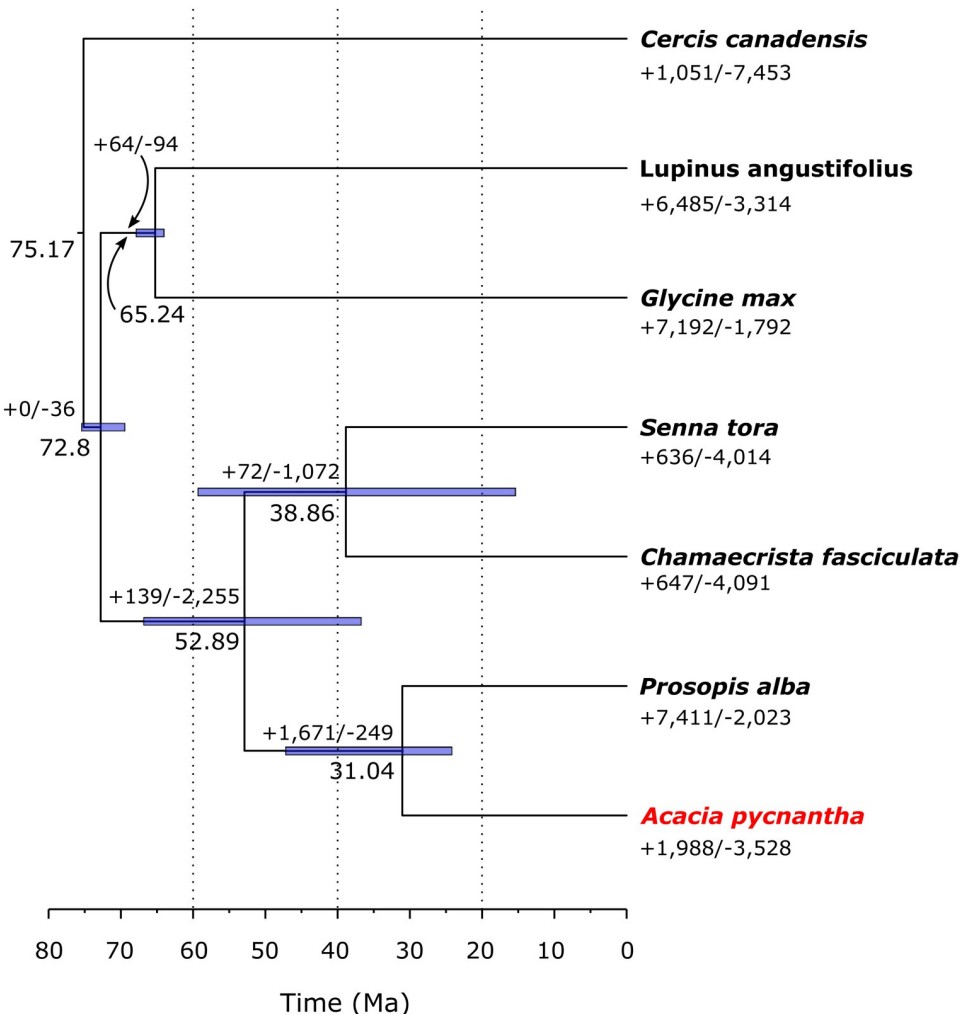

**Fig 4. A Leguminosae-only chronogram, with expanded and contracted gene families estimated by CAFÉ.**
Numbers above nodes represent median divergence age estimates. Gene family expansions (+) and contractions (-) are
shown below the nodes. Error bars represent 95% posterior probability estimates of divergence times.

in S1 File). Anchor-pair analyses for both *Acacia* and *Prosopis* recovered a peak at $K_S$ ~0.8 (Fig
5), suggesting a shared duplication event for these two Caesalpinioideae. *Senna*, another Cae-
salpinioideae, shows an anchor-pair peak at $K_S$ ~0.6; it is difficult to tell whether these three
peaks represent a WGD event shared by the three Caesalpinioideae (with the slightly lower $K_S$
peak in *Senna* caused by differing evolutionary histories and selection on synonymous codon
positions), or two separate WGD events. In either case, these WGD events likely occurred
shortly after the common ancestor of *Acacia* and *Lupinus* (the latter belonging to the Papilio-
noideae) diverged, because the *Acacia-Lupinus* speciation peak occurs at $K_S$ ~0.8, while the
*Lupinus* anchor-pair analyses did not recover a WGD peak at a similar position.

A duplication event occurring at or near the divergence between the subfamilies Caesalpi-
nioideae and Papilionoideae has previously been identified using transcriptome phyloge-
nomics [80, 81]. Evidence for a Caesalpinioideae-specific polyploidy event was identified in
Cannon et al. (2015) and Zhao et al. [82] but not by Koenen et al. (2021), who instead deter-
mined that a WGD was shared by Caesalpinioideae and Papilionoideae before the two

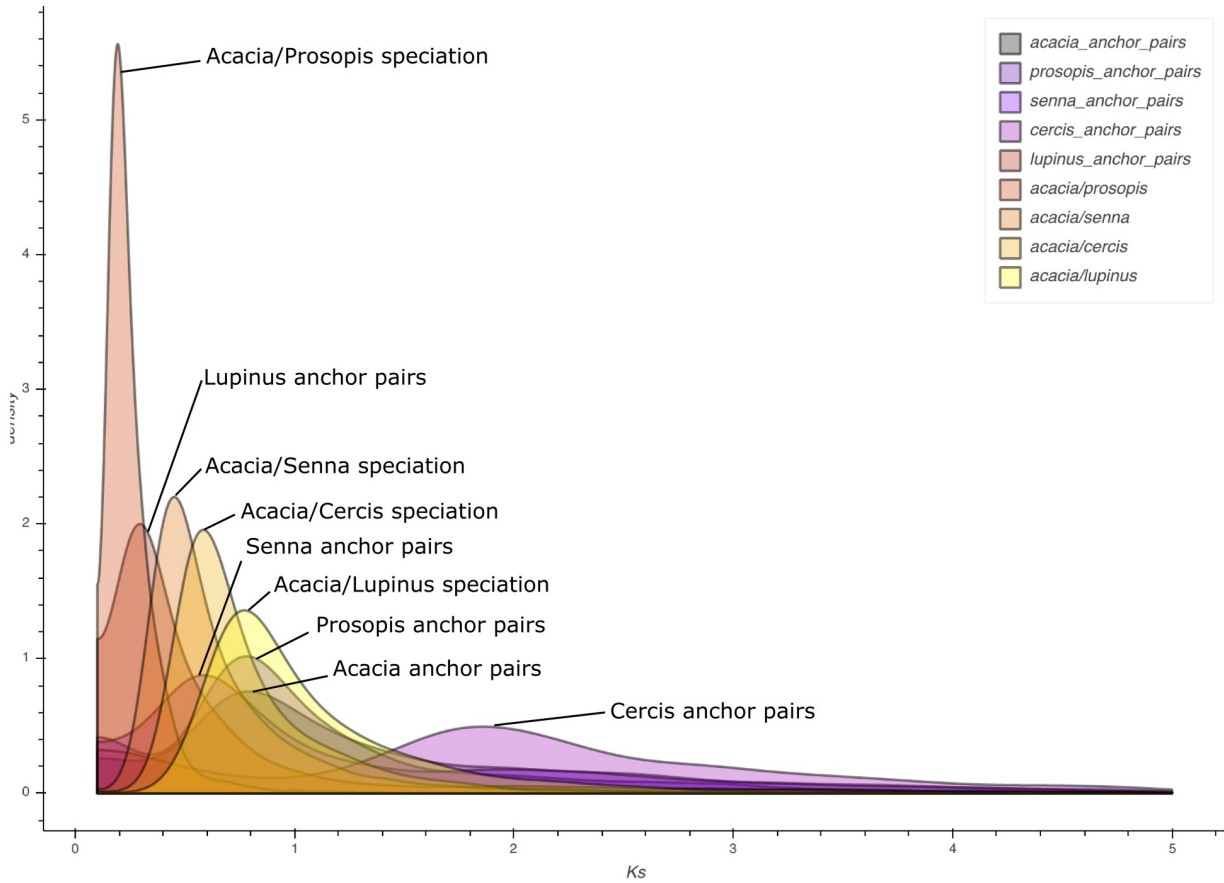

**Fig 5. Kernel Density Estimate (KDE) plot of K$_s$ distributions from one-to-one orthologs for *Acacia pycnantha* vs other Leguminosae taxa, and from anchor-pair paralogs for *Acacia pycnantha* and other Leguminosae taxa.** Peaks for one-to-one orthologs provides a proxy for the relative speciation time for each taxon pair, whereas peaks for anchor-pair paralogs can provide a proxy for the relative time of gene or putative whole genome duplication for each taxon.

subfamilies diverged. We also detected no recent WGD events in *Cercis*, a unique member of the Leguminosae that has been previously found to have no signal of a recent genomic duplication, unlike the rest of the family [83]. The peak in the *Cercis* anchor-pair plot at ~1.9 potentially reflects the gamma duplication shared by all eudicots [84], but this begs the question of why this duplication event was not detected for any of the other taxa. Greater taxonomic sampling of genomic data, especially from the non-Papilionoideae subfamilies of Leguminosae, will be crucial to resolve questions pertaining to the timing and placement of duplications in the evolutionary history of the family.

Analyses in this study used a fixed topology with *Cercis* as sister to the Caesalpinioideae+Papilionoideae, based on Leguminosae relationships recovered from previous large-scale phylogenetic analyses [12, 52, 83]. However, ML analyses based on the 85 single copy ortholog (SCO) concatenated alignments in this study produced a topology with *Cercis* as sister to the Caesalpinioids (73% UFBoot, 68.6% SH-LRT), with Papilionoids branching earlier (this was also the topology recovered using STAG [85] within OrthoFinder). Gene concordance factors calculated from the 85 SCOs indicate that there is disagreement among individual gene trees regarding the position of *Cercis*, with 32/85 genes resolving it as sister to the Caesalpinioids, 15/85 resolving it as sister to Papilionoids, and 31/85 resolving it as sister to Caesalpinioids+Papilionoids (S1G, S7 Fig in S1 File). The WGD analyses suggest that the split between *Acacia*

and *Cercis* is more recent than that between *Acacia* and *Lupinus*, providing another example of contentious placement of the Cercidoideae as sister to the rest of Leguminosae (Fig 5, S1F, S5-S7 Figs in S1 File). Our SCO analyses are based on limited taxon sampling, which may explain conflicting placements in the phylogenetic results, as we are missing key lineages that may help resolve the position of *Cercis*. However, additional taxon sampling would not change the positions of the divergence peaks based on $K_S$ analysis as it is performed on species-pairs. Additionally, the results from $K_S$ analysis agree with approximately 40% of the SCO phylogenies. Literature exploring differences in divergence patterns between $K_S$ analysis and phylogenetic analyses is lacking and this topic is worth further research. Inferring the sequence of divergence events compared to whole genome duplications using $K_S$ plots is sometimes unreliable due to variation in synonymous substitution rates between the lineages involved [86]. This could be addressed using the more complex models of evolution employed in modern phylogenetics. Uncertainty in the relationships among these subfamilies is not unexpected, as studies with the most comprehensive sampling to date in terms of both taxa and loci [52] found conflicting signal in the backbone of the Leguminosae and the branching order of the subfamilies. The rapid radiation of Leguminosae subfamilies [81], and the fact that *Cercis* has not undergone any genome duplications may be contributing to the conflict. The position of *Cercis* in the Leguminosae has implications for our understanding of evolution and classification of the legumes, including identifying polyploidy events in the family.

## Comparative genomics

The radiation of *Acacia* has a broad ecological amplitude, from wet forest to the arid zone, through a wide range of geological substrates, and has occurred relatively recently (ca. 23 Ma), featuring a staggering diversity in habit, vegetative/photosynthetic organs, and reproductive organs [5]. To investigate gene families that have expanded or contracted in *A. pycnantha* at a significant rate, we generated OrthoFinder orthogroups using Leguminosae proteomes only, and performed rate analyses using CAFÉ. A chronogram showing the number of gene family expansions and contractions at each node is shown in Fig 4. Of the 2,415 expanded gene families in *A. pycnantha*, 40 were predicted to be evolving at significantly elevated rates, whereas 26 of the 2,331 contracted gene families were predicted to be evolving at significantly elevated rates. These gene families that expanded rapidly in *A. pycnantha* were further explored by identifying significantly enriched GO terms and KEGG orthologs (S24-S26 Tables in S2 File). Enriched GO terms included functions associated with DNA repair and telomere maintenance, binding of metal ions (including zinc, magnesium, calcium, and iron), and defence responses. Enriched KEGG pathways showed significant enrichment for genes involved in stress management, hormone signalling and carbohydrate metabolism pathways (S26 Table in S2 File). Expansion and contraction of gene families is considered important in adaptive diversification [87], and investigating enriched GO terms and KEGG pathways in functional studies can provide insights into the evolutionary adaptations of organisms.

To further examine putative functions of expanded and contracted gene groups in *A. pycnantha*, we investigated Pfam protein domains that were highly enriched or reduced in comparison to the average number in the 15 other angiosperms included in this study. In *A. pycnantha*, 193 Pfam domains were significantly enriched. GO enrichment analyses of genes containing these Pfam domains recovered GO terms associated with cell-wall development (trehalose metabolic process, xyloglycan metabolic process, cellulose biosynthetic process), transmembrane transport, and phosphatase activity (S27, S28 Tables in S2 File). KEGG enrichment analyses of expanded PFAM domains include pathways associated with diterpenoid biosynthesis and carbon fixation (S29 Table in S2 File).

Interestingly, one of the most expanded Pfam domains in *Acacia* relative to the other genomes was PF07985.13, which was annotated as "SRR1/Protein SENSITIVITY TO RED LIGHT REDUCED". *Acacia* has 17 copies of this domain present in 15 genes; two genes contain two copies of the domain, and no genes contain this domain and another domain type; other angiosperm genomes had 1–3 copies. Phylogenetic analysis of the orthogroups associated with this domain recovered the topology as expected for the angiosperm phylogeny, and the *Acacia* and *Prosopis* SRR1 sequences occur in two sister clades (Fig 6A). One clade has two copies of *Acacia* SRR1 genes and one copy of *Prosopis* SRR1 (Clade 1). The other clade includes 16 copies of *Acacia* SRR1 sequence, and one copy from *Prosopis* (Clade 2). There are multiple subclades of *Acacia* SRR1 proteins in Clade 2, and the different copies of the SRR1 annotated genes in Clade 2 occur on long branches relative to the rest of the phylogeny indicating extensive sequence divergence between clades and gene copies. The SRR1 domain (Fig 6B) reflects some of this sequence divergence in *Acacia*, especially between amino acid positions 10–22, though there are three sections of highly conserved amino acid sequences across all angiosperms. Much of the sequence variability in *Acacia* SRR1 genes occurs outside the predicted SRR1 domain (S6 File).

SRR1 is well-characterised in *Arabidopsis thaliana* and is involved in light-signalling via phytochrome B and regulation of circadian rhythms. SRR1 null mutants have early flowering phenotypes. SRR1 regulates several transcription factors that are repressors of Flowering Time (FT) and acts as an integrator between photoperiodic regulation and other pathways to maintain repression of flowering in unsuitable conditions [88]. *Acacia* species are known to have fine-tuned flowering times, with highly synchronous flowering events across many different species occurring in early spring. Glasshouse experiments have shown that *A. pycnantha* produces flower buds year-round [89] and flowering is triggered by environmental conditions such as temperature and rainfall [90]. Diversification of a gene associated with repression of flowering time except under ideal conditions could be linked to the strong pattern of regular, synchronous flowering in many species of *Acacia*.

Finally, we investigated enrichment of GO terms an KEGG pathways in *Acacia*-specific orthogroups and unigenes. Of the 15 angiosperm taxa examined, *Acacia* had the highest number of species-specific orthogroups (1,759 orthogroups, containing 7,207 genes), and a high number of genes that were not assigned to orthogroups (unigenes = 4,091). For *Acacia*-only orthogroups, enriched GO terms included functions corresponding to oxidoreductase activity, transcription regulation, and programmed cell death (S30, S31 Tables in S2 File); enriched KEGG pathways related to circadian rhythm and cell death signalling (S32 Table in S2 File). For *Acacia* unigenes, enriched GO terms included functions associated with nitrogen utilisation and metabolism, phosphatase activity (S33, S34 Tables in S2 File); enriched KEGG pathways related to ribosome structure and development (S35 Table in S2 File). Species-specific genes tend not to include basic genes for plant development and function, such as those relating to plant structure or photosynthesis [91]. Rather, they can be involved in functions that are important for adaptation to specific environmental or evolutionary conditions and represent unique traits of a species [92, 93]. Functional investigations of these *Acacia*-specific genes may yield insights into the evolutionary success of *Acacia* in Australia.

## Conclusion

In this study, we assembled a draft genome of *Acacia pycnantha*, comprising 1,267 scaffolds with an N50 of ~2.8 Mb, and totaling ~0.8144 Gb in length. The annotated genome includes 47,624 genes, of which 94% were functionally annotated; 62% of the genome was determined to be transposable elements. We also developed a method to identify and characterize plastid

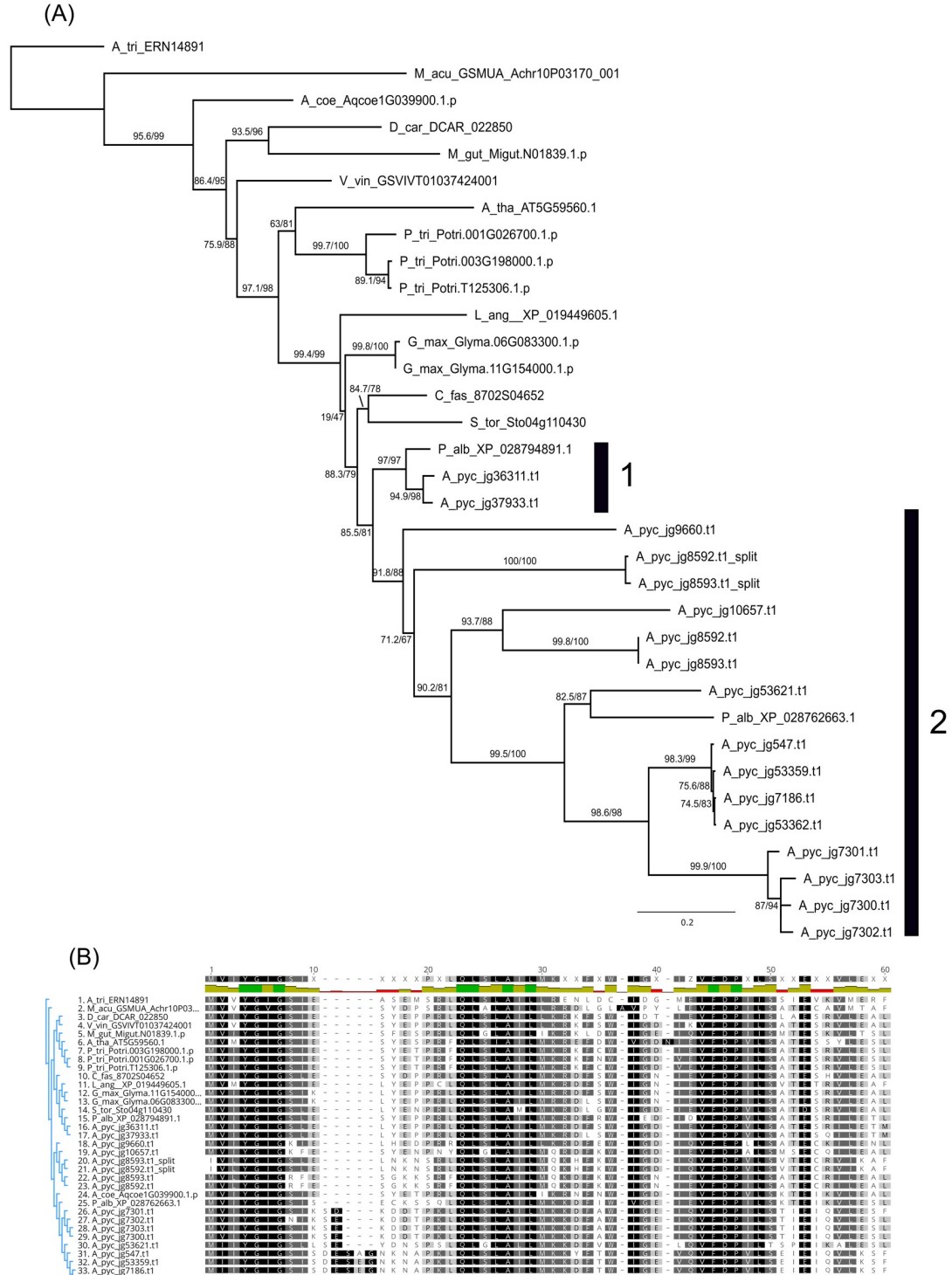

**Fig 6. SRR1 diversity and phylogenetic relationships in sampled angiosperms.** (A) Phylogenetic analysis of amino acid genes sequences with an SRR1 domain. Numbers at nodes represent SH-aLRT values/UFBoot support values. (B) Alignment of the SRR1 domain identified by Pfam, shaded by similarity (darker = more similar).

or mitochondrial transfers to the nuclear genome and confirmed over 800 kb of such transfers in the *A. pycnantha* genome. Phylogenetic dating indicated a divergence between *Acacia* and *Prosopis* approximately 24 to 47 Ma, and analysis identified a whole genome duplication either at the base of Caesalpinioideae, or at the time of divergence between Caesalpinioideae and Papilionoideae. Concordance factor analysis of 85 single-copy orthologs, and KDE plots of Ks distributions in the Leguminosae indicated conflict in the relationships between the subfamilies. Investigation of gene family expansions, both with CAFÉ analyses and Pfam z-scores, and subsequent analysis of GO term enrichment and KEGG pathway enrichment of expanded families, indicated a suite of putative genes important in the evolution and diversification of *Acacia*. This genome provides a valuable resource for a wide range of questions regarding *Acacia* evolution, genetics, forestry, and ecology.

## Supporting information

**S1 File. Additional bioinformatic methods and results.** S1 A. NECAT assembly configuration details; file <acacia_config.txt>; S1 B. Merqury results and spectra plot; S1 C. Fixed topology trees used for OrthoFinder runs; S1 D. Visualisation of PhyloBayes chain_1.trace file in Tracer, showing log likelihood; S1 E. CAFÉ methods; S1 F. Whole Genome Duplication KDE plots; S1 G. Gene tree concordance factors of Leguminosae.
(DOCX)

**S2 File. Supporting results from assembly, analyses, and annotations (S1-S34 Tables).**
(XLSX)

**S3 File. Output of InterProScan annotation of predicted gene set.**
(ZIP)

**S4 File. PhyloBayes input alignments.**
(ZIP)

**S5 File. GFF file for NUPT, NUMT, and NUMPT detected in *A. pycnantha* genome.**
(ZIP)

**S6 File. SRR1 alignment of whole predicted gene region (FASTA format).**
(FASTA)

## Acknowledgments

We gratefully acknowledge Mabel Lum (Bioplatforms Australia) for coordinating sample submission and sequencing, Tamera Beath (Australian National Botanic Gardens) for supporting tissue collection from the golden wattle plant, Dave Marshall (CSIRO) for performing flow cytometry analysis, and Ashley Jones (Australian National University) for advice on DNA extractions for ONT sequencing. The Genomics of Australian Plants consortium is acknowledged for funding.

## Author Contributions

**Conceptualization:** Todd G. B. McLay, Daniel J. Murphy, Sarah Mathews, Chris J. Jackson.

**Data curation:** Theodore R. Allnutt, Chris J. Jackson.

**Formal analysis:** Todd G. B. McLay, Chris J. Jackson.

**Funding acquisition:** Todd G. B. McLay, Daniel J. Murphy, Sarah Mathews, Gillian K. Brown.

**Investigation:** Todd G. B. McLay, Gareth D. Holmes, Theodore R. Allnutt, Chris J. Jackson.

**Methodology:** Todd G. B. McLay, Theodore R. Allnutt, Chris J. Jackson.

**Project administration:** Todd G. B. McLay.

**Software:** Theodore R. Allnutt, Chris J. Jackson.

**Supervision:** David J. Cantrill, Frank Udovicic.

**Validation:** Todd G. B. McLay.

**Visualization:** Todd G. B. McLay, Chris J. Jackson.

**Writing – original draft:** Todd G. B. McLay, Daniel J. Murphy, Sarah Mathews, Chris J. Jackson.

**Writing – review & editing:** Todd G. B. McLay, Daniel J. Murphy, Gareth D. Holmes, Gillian K. Brown, David J. Cantrill, Frank Udovicic, Theodore R. Allnutt, Chris J. Jackson.

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
