## [Decision Letter · Decision Letter 0]

30 Jun 2022

PONE-D-22-13018A genome resource for Acacia, Australia’s largest plant genusPLOS ONE

Dear Dr. McLay,

Thank you for submitting your manuscript to PLOS ONE. After careful consideration, we feel that it has merit but does not fully meet PLOS ONE’s publication criteria as it currently stands. Therefore, we invite you to submit a revised version of the manuscript that addresses the points raised during the review process. Both the reviewers gave a fully positive review of the MS, suggesting a few minor changes.

We look forward to receiving your revised manuscript.

Kind regards,

Serena Aceto, Ph.D.

Academic Editor

PLOS ONE

Journal Requirements:

Reviewers' comments:

Reviewer's Responses to Questions

**Comments to the Author**

1. Is the manuscript technically sound, and do the data support the conclusions?

Reviewer #1: Yes

Reviewer #2: Yes

2. Has the statistical analysis been performed appropriately and rigorously? 

Reviewer #1: Yes

Reviewer #2: Yes

3. Have the authors made all data underlying the findings in their manuscript fully available?

Reviewer #1: Yes

Reviewer #2: Yes

4. Is the manuscript presented in an intelligible fashion and written in standard English?

Reviewer #1: Yes

Reviewer #2: Yes

5. Review Comments to the Author

Reviewer #1: Manuscript PONE-D-22-13018 presents a reference genome for the plant genus Acacia. The article is well-motivated, technically sound, and is presented in an interesting evolutionary context. I think it is suitable for publication in PLoS One in essentially its current form, and beyond that, will become a highly useful molecular resource.

I have a small number of very minor comments for the authors:

Were any long-read assemblers besides NECAT tested with these data? If such tests were undertaken, and NECAT seemed like the best assembly, I would encourage the authors to share relevant observations in a supplement, since they might help guide the choices of researchers with similar long read datasets. Note, however, I will understand if the authors decline to do this. One sometimes tests assemblers only enough to determine that they are unlikely to produce the best assembly, and without optimizing enough for the purposes of a fair comparison… If so, understandable if they authors don’t want to share outcomes of relatively superficial tests of a package. However, if the authors did generate assemblies with other software somewhat carefully, the results are potentially interesting to many readers.

Figure captions do not seem to be visible in the version of the manuscript I downloaded, so I have not had an opportunity to review these. In particular, I think this led to some difficult interpreting Figure 3, especially the bottom two panels. For example, in the figure on the left, I wondered what the on-diagonal values meant – the number of orthogroups that an assembly shares with itself (is this the total number of genes, or some subset used in orthogroup analysis)? I also wondered if some normalization of these values could offer an interesting perspective? For example, rather than the numbers of genes, the proportion of genes in one species with orthologues in another…?

Line 544 "a broad"?

Reviewer #2: McLay et al. reported their study on genome assembly and characterization of a Acacia species. Acacia genus and the studied species has significant values in many aspects. High quality genome assembly, annotation and characterization were generated and presented. Analytical methods, interpretation and discussion are all well.

Accept with a minor revision.

minor comments:

1. line 287-28, in Abstract, please improve the confusing the English expression. You know, we can not generate long-read sequences using Illumina squences.

"We generated long-read sequences for A. pycnantha using Oxford Nanopore, 28 10x Genomics Chromium linked reads, and short-read Illumina sequences,"

2. line 68, "(e.g. [10]", sth lost here, at least one ")".

3. line 346, I saw "Gbp", in line 347, I saw "Gb". Please follow a constant. Also, I read both "Kbp" and "Kb". Take care of them.

4. line 356, "an N50 of 0.963 Gb"? please make it sure.

5. Table 1. Do not use Captial letters for all words. And please provide units for some items such as "Contig N50".

6. line 389-392, confusing sentence. "BLAST % identity" is read not a formal English expression.

6. PLOS authors have the option to publish the peer review history of their article (what does this mean?). If published, this will include your full peer review and any attached files.

Reviewer #1: No

Reviewer #2: No

---

## [Author Response · Author response to Decision Letter 0]

1 Aug 2022

Dear Dr. Aceto,

Please find below a statement of the revisions performed on the manuscript, based on the comments of the two reviewers and yourself. Thank you for your efforts helping our manuscript get published. 

Regards,

Todd McLay

Manuscript checked for style and formatting. 

Funding from the BioPlatforms – Genomics for Australian Plants is not a grant but a cofounding opportunity, and as such there is no grant number. This is stated in the in-text Funding Information but I could not change the financial disclosure statement. It should read 'No specific funding was obtained for this study'. 

The genome and associated data is under the existing BioProject accession PRJNA752212. 

This is now in the correct place. 

5. Review Comments to the Author

Reviewer #1: Manuscript PONE-D-22-13018 presents a reference genome for the plant genus Acacia. The article is well-motivated, technically sound, and is presented in an interesting evolutionary context. I think it is suitable for publication in PLoS One in essentially its current form, and beyond that, will become a highly useful molecular resource.

I have a small number of very minor comments for the authors:

Were any long-read assemblers besides NECAT tested with these data? If such tests were undertaken, and NECAT seemed like the best assembly, I would encourage the authors to share relevant observations in a supplement, since they might help guide the choices of researchers with similar long read datasets. Note, however, I will understand if the authors decline to do this. One sometimes tests assemblers only enough to determine that they are unlikely to produce the best assembly, and without optimizing enough for the purposes of a fair comparison… If so, understandable if they authors don’t want to share outcomes of relatively superficial tests of a package. However, if the authors did generate assemblies with other software somewhat carefully, the results are potentially interesting to many readers.

We have added in information about the other assemblers we tested in the main text, and the results of those tests as Supp. Information File S2:Table S4. 

Figure captions do not seem to be visible in the version of the manuscript I downloaded, so I have not had an opportunity to review these. In particular, I think this led to some difficult interpreting Figure 3, especially the bottom two panels. For example, in the figure on the left, I wondered what the on-diagonal values meant – the number of orthogroups that an assembly shares with itself (is this the total number of genes, or some subset used in orthogroup analysis)? I also wondered if some normalization of these values could offer an interesting perspective? For example, rather than the numbers of genes, the proportion of genes in one species with orthologues in another…?

Figure captions were in body of manuscript as per PLoS submission guidelines. These have been slightly modified in some cases for accuracy. For figure 3C (heatmap visualisations of orthogroup and ortholog species comparisons), we would prefer to keep the current visualisation type (i.e. raw numbers), rather than normalising, as we believe these numbers provide useful insights into the genomes of each taxon. However, to better visualise these values in the left heatmap (orthogroups), we have rescaled the colour values so that the smallest value in our analysis corresponds to the darkest colour – this provides a greater range of values that can be visualised, and we hope makes the patterns much clearer. The corresponding code has been updated in the manuscript git repository. We also agree that the meaning of the on-diagonal values in both heatmaps were a bit obtuse, and we have added an explanation to the figure legend. 

Line 544 "a broad"?

Added space as per reviewer. 

Reviewer #2: McLay et al. reported their study on genome assembly and characterization of a Acacia species. Acacia genus and the studied species has significant values in many aspects. High quality genome assembly, annotation and characterization were generated and presented. Analytical methods, interpretation and discussion are all well.

Accept with a minor revision.

minor comments:

1. line 287-28, in Abstract, please improve the confusing the English expression. You know, we can not generate long-read sequences using Illumina squences.

"We generated long-read sequences for A. pycnantha using Oxford Nanopore, 28 10x Genomics Chromium linked reads, and short-read Illumina sequences,"

Changed sentence structure to improve clarity. 

2. line 68, "(e.g. [10]", sth lost here, at least one ")".

Removed the ‘e.g.’, so now the sentence ends with the reference. 

3. line 346, I saw "Gbp", in line 347, I saw "Gb". Please follow a constant. Also, I read both "Kbp" and "Kb". Take care of them.

Changed all occurrences to Gb and kb. 

4. line 356, "an N50 of 0.963 Gb"? please make it sure.

Changed Gb to kb.

5. Table 1. Do not use Captial letters for all words. And please provide units for some items such as "Contig N50".

Changed the font and added units for terms. 

6. line 389-392, confusing sentence. "BLAST % identity" is read not a formal English expression.

Changed to BLAST percent identity. 

Additional changes

1. Figure 3 and Figure 6 have had panel labels added/changed. Figure 3 was changed so the scale on the heatmap is more easily interpreted (no changes to data inputs). 

a. Fig3-orthofinder_figure-revisedJuly2022.eps

b. Fig6-SRR1_phylogeny_PFAMdomain-revisedJuly2022.eps

2. A new supporting information file has been uploaded including a table addressing the comments of R1 regarding other genome assembly softwares used

a. Fig6-SRR1_phylogeny_PFAMdomain-revisedJuly2022.xlsx

3. Reference list has been checked more closely for accuracy of references, no references were removed, references for assemblers have been added. 

4. Position of supporting information details moved to end of manuscript, after reference list.

---

## [Decision Letter · Decision Letter 1]

25 Aug 2022

A genome resource for Acacia, Australia’s largest plant genus

PONE-D-22-13018R1

Dear Dr. McLay,

We’re pleased to inform you that your manuscript has been judged scientifically suitable for publication and will be formally accepted for publication once it meets all outstanding technical requirements.

Kind regards,

Serena Aceto, Ph.D.

Academic Editor

PLOS ONE

Additional Editor Comments (optional):

Reviewers' comments:

Reviewer's Responses to Questions

**Comments to the Author**

1. If the authors have adequately addressed your comments raised in a previous round of review and you feel that this manuscript is now acceptable for publication, you may indicate that here to bypass the “Comments to the Author” section, enter your conflict of interest statement in the “Confidential to Editor” section, and submit your "Accept" recommendation.

Reviewer #1: All comments have been addressed

Reviewer #2: All comments have been addressed

2. Is the manuscript technically sound, and do the data support the conclusions?

Reviewer #1: Yes

Reviewer #2: Yes

3. Has the statistical analysis been performed appropriately and rigorously? 

Reviewer #1: Yes

Reviewer #2: Yes

4. Have the authors made all data underlying the findings in their manuscript fully available?

Reviewer #1: Yes

Reviewer #2: Yes

5. Is the manuscript presented in an intelligible fashion and written in standard English?

Reviewer #1: Yes

Reviewer #2: Yes

6. Review Comments to the Author

Reviewer #1: My previous comments have been adequately addressed. Congratulations to the authors on a fine article.

Reviewer #2: This is the revision from McLay et al. and reports the study on genome assembly and characterization of a Acacia species. This revision satisfied all my comments. No further comments.

7. PLOS authors have the option to publish the peer review history of their article (what does this mean?). If published, this will include your full peer review and any attached files.

Reviewer #1: No

Reviewer #2: No

---

## [Editor Report · Acceptance letter]

6 Oct 2022

PONE-D-22-13018R1 

A genome resource for *Acacia*, Australia’s largest plant genus 

Dear Dr. McLay:

I'm pleased to inform you that your manuscript has been deemed suitable for publication in PLOS ONE. Congratulations! Your manuscript is now with our production department. 

Kind regards, 

on behalf of

Dr Serena Aceto 

Academic Editor

PLOS ONE